# Life History Effects Linked to an Advantage for *w*Au *Wolbachia* in *Drosophila*

**DOI:** 10.3390/insects10050126

**Published:** 2019-05-02

**Authors:** Li-Jun Cao, Weibin Jiang, Ary A. Hoffmann

**Affiliations:** 1Pest and Adaptation Research Group, School of BioSciences, Bio21 Institute, The University of Melbourne, Melbourne, VIC 3010, Australia; jiangwb@shnu.edu.cn; 2Institute of Plant and Environmental Protection, Beijing Academy of Agriculture and Forestry Sciences, Beijing 100097, China; 3College of Life & Environmental Science, Shanghai Normal University, Shanghai 200234, China

**Keywords:** *Wolbachia*, *Drosophila*, fitness advantage, endosymbionts

## Abstract

*Wolbachia* endosymbiont infections can persist and spread in insect populations without causing apparent effects on reproduction of their insect hosts, but the mechanisms involved are largely unknown. Here, we test for fitness effects of the *w*Au infection of *Drosophila simulans* by comparing multiple infected and uninfected polymorphic isofemale lines derived from nature. We show a fitness advantage (higher offspring number) for lines with the *w*Au *Wolbachia* infection when breeding on grapes, but only where there was *Talaromyces* and *Penicillium* fungal mycelial growth. When breeding on laboratory medium, the *w*Au infection extended the development time and resulted in larger females with higher fecundity, life history traits, which may increase fitness. A chemical associated with the fungi (ochratoxin A) did not specifically alter the fitness of *w*Au-infected larvae, which developed slower and emerged with a greater weight regardless of toxin levels. These findings suggest that the fitness benefits of *Wolbachia* in natural populations may reflect life history changes that are advantageous under particular circumstances, such as when breeding occurs in rotting fruit covered by abundant mycelial growth.

## 1. Introduction

Endosymbionts can have a variety of fitness effects on their invertebrate hosts, many of which are beneficial to the hosts, particularly through the provision of nutritional resources. For instance, in tsetse flies, *Wigglesworthia glossinidia* provides nutrients essential for female fertility [1], while in aphids, *Buchnera aphidicola* synthesizes essential amino acids [2]. Similarly, in ticks, *Francisella* bacteria synthesize B vitamins, which are lacking in their blood meals [3]. Not all invertebrate endosymbionts necessarily have beneficial effects on their hosts. For instance, aphids carry a pea aphid secondary symbiont (PASS), which suppresses growth and reproduction when *Buchnera* symbionts are present, although PASS also benefits aphids in the absence of *Buchnera* [4].

Of the different endosymbionts found in insects and other invertebrates, *Wolbachia* infections are the most widely studied because of their widespread distribution across invertebrates [5,6]; their diverse effects on host fitness, including various reproductive effects [7]; and their capacity to interfere with viral transmission in invertebrate hosts and thereby impact on human diseases associated with arboviruses [8,9,10,11]. *Wolbachia* strains can affect the reproduction of hosts in such a way as to enhance their spread and persistence in natural populations. This is particularly associated with cytoplasmic incompatibility (CI), the reduction in embryo and offspring viability when females that lack *Wolbachia* mate with males that carry *Wolbachia*, ensuring a fitness advantage of infected females over uninfected females [12] and leading to a rapid spread of *Wolbachia* in natural populations [13].

Although much of the focus of *Wolbachia* research has been on infections that spread via CI or that cause other reproductive effects, like feminization and male killing, many *Wolbachia* infections from natural populations also provide a range of benefits to their hosts [6]. This includes nutritional benefits in bedbugs [14] as well as the mutualistic associations of *Wolbachia* in filarial nematodes [15]. In *Drosophila*, *Wolbachia* may also provide nutritional benefits [16] although fitness effects on hosts often remain unclear and have only been identified indirectly [17,18].

One of the first *Wolbachia* infections to be described with uncertain host effects was the *w*Au infection of *Drosophila simulans*, identified from natural Australian populations [18], where it showed a potential to increase in frequency even though it had no impact on CI or other host reproductive effects [19]. Recent population cage experiments where flies were raised on fermenting fruits indicated a substantial fitness advantage of the *w*Au infection of around 20% per generation [20]. This infection is particularly interesting because after transfer to *Aedes* mosquitoes it has the potential to strongly inhibit Zika and dengue virus transmission [8]. If the *w*Au infection is associated with positive effects on fitness under some situations, this might assist in its spread from a low frequency in native hosts, which can normally be difficult to achieve when *Wolbachia* strains only influence host reproduction [21,22].

In this paper, we use a novel design to investigate the fitness advantages that might be associated with the *w*Au infection in *Drosophila simulans* by undertaking experiments that build on earlier work [19,20], which pointed to fitness advantages of this infection, but without identifying the relevant causes. We initially focused on the fitness of flies carrying *w*Au breeding in rotting fruit with different levels of *Talaromyces* and *Penicillium* fungal growth. We test whether a common compound associated with the predominant fungi (ochratoxin A) might mediate the fitness advantage of *w*Au in these breeding sites. However, we instead find that *w*Au-infected larvae had a delayed development time, but larger emergence size regardless of breeding conditions, and we suspect these changes are critical for the infection to persist and spread in sites with abundant fungal growth. Thus, an overall life history impact of this *Wolbachia* infection that does not affect host reproduction seems to form the basis for a fitness advantage.

## 2. Methods

### 2.1. Lines and Infection Status

Multiple lines of *D. simulans* (70 infected, 70 uninfected) were established for these experiments, each from 20 to 30 offspring of different field-collected females. Each isofemale line was confirmed to be either *w*Au-infected or uninfected based on a molecular assay (below). The lines were initiated with the offspring of field-caught females collected at the same time around Roleystone near Perth, Western Australia, where the infection frequency was around 60% based on data collected from the current study. Offspring of the field females forming a line were reared at a low density in 150 mL vials with a dead yeast-treacle- agar fly medium [19] for two generations before the three experiments (Table 1) started. Only lines reared on this medium were used in experiments. Isofemale lines were scored again for *Wolbachia* status after the experiments to confirm they were infected.

Flies were assayed for *Wolbachia* infection status and strain type using a real-time PCR/high resolution melt (RT-PCR/HRM) method designed to amplify a fragment of the *wsp* gene following Chelex extraction of DNA from individual flies using *wsp* primers (*wsp_*validation_Fwd: 5′-TTGGTTACAAAATGGACGACATCAG-3′; *wsp_*validation_Rev: 5′-CGAAATAACGAGCTCCAGCATAAAG-3′) [19,23]. These primers produce different melting temperatures of the *wsp* products. Separate *D. simulans* specific primers (Dsim RpS6 Fwd: 5′-CCAGATCGCTTCCAAGGAGGCTGCT-3′; DsimRpS6 Rev: 5′-GCCTCCTCGCGCTTGGCCTTAGAT-3′) were used as an internal control to test for the quality of the DNA extraction procedure, confirm species identification, and ensure that the correct PCR conditions were present. Individual flies from *w*Au-infected and *w*Ri-infected *D. simulans* mass bred laboratory lines were used as positive and strain type controls for each PCR plate, while individual flies from an uninfected mass bred line and “no template” PCR reactions were used as negative controls. This assay can distinguish *w*Au from *w*Ri infected individuals as well as uninfected individuals, which is critical because the *w*Ri infection has recently spread through *D. simulans* populations in eastern Australia where only the *w*Au infection had previously been detected [19]. The *w*Ri infection has not yet been found in Western Australia [20].

### 2.2. Comparison of Lines on Grapes with Fungal Growth

Kriesner and Hoffmann [20] showed that *w*Au had a fitness advantage when flies were held in population cages that were maintained on fruit, and that conditions provided by fruit may have accounted for the substantial fitness advantage of *w*Au, which resulted in an increase in infection frequency. We tested this by using organic table grapes (Red Globe) not contaminated by pesticides. Pairs of male and female adults (from each of the 23 *w*Au-infected and 23 uninfected lines, 3 pairs per isofemale line) were set up when *D. simulans* were five days post eclosion.

Adults were provided with bottles (250 mL) containing grapes, which were cut in half and placed on vermiculite. To accustom the flies to the grapes and conditions, flies were initially held on one grape per bottle (cut in half) for three days. Bottles were placed in a shade house, where they were exposed to fluctuating ambient conditions (temperatures in ranged from 13.9 to 31.6 °C, natural daylight). Adult pairs were then transferred (or “tipped”) to new bottles after three days of conditioning on the initial grapes. There were 6 grape halves (rather than 2) per bottle in these new bottles to ensure the fruit was not limiting for offspring production. Adults were left in these bottles for 3 days and then transferred to a new set of bottles with the same amount of grape resource for an additional three days. Emerging numbers of adults in bottles from this second and third set of bottles (offspring per pair of adults) were recorded after collecting flies 18 days after bottles had been set up (about 3 days after the first flies were observed). Where any male or female parent had died in a bottle (14.1% of cases), bottles were not used in life history assessments.

We suspected that growth of fungi, bacteria, or yeast might be an important factor influencing the fitness of the *Wolbachia* infection. We therefore compared the offspring number between *w*Au-infected and uninfected lines, focusing on the amount of fungal growth, which was easy to assess visually through the amount of mycelial mass. Fungus infestation levels on grapes were evaluated on the 18th day after flies were set up, with 3 levels recognized: Level 1, less than 1 piece of 6 pieces of grapes was covered by visible mycelial growth; level 2, 2 to 4 pieces of grapes were covered by visible growth; level 3, 5 to 6 pieces of grapes were covered by visible growth (Appendix A). We compared the offspring number between *w*Au-infected and uninfected lines (infection as a fixed effect) with ANOVAs (IBM SPSS Statistics version 25) that also included fungus level as a fixed effect after testing for normality with a Kolmogorov-Smirnov test. Note that there were at least 9 replicate bottles (and a maximum of 38 bottles) in an infection-fungal growth level category. Because there were interactions between the fungal category and whether or not lines were infected (see Results below), we further considered the effect of the infection at each fungal level with t tests and showed patterns with kernel density (violin) plots, which show how trait values are distributed (R software package *vioplot*
https://cran.r-project.org/web/packages/vioplot/index.html).

To examine fly weight, females and males from the same bottle were separated into glass vials and dried at 50 °C for 24 h in an oven. Weight of females and males from each bottle was obtained using an analytical balance (Sartorius Type 1712, Sartorius AG, Gottingen, Germany) with accuracy up to 0.01 mg. We excluded bottles with fewer than 4 individuals. Average weight of females and males from each bottle (expressed as average weight per fly in a bottle) were compared between *w*Au-infected and uninfected lines with an ANCOVA, which included adult parental age (successive tips) and fungal growth as fixed effects; emergence number was included as a covariate because adult weight may depend on larval density, although this effect was not significant. Sex ratio (female/(female + male)) was estimated by adding up all the offspring produced by a single pair of flies averaged across the two observation periods. A nested factor due to the isofemale line was included in initial models for this experiment and the other experiments did not appreciably affect the significance of infection effects, which is the focus of the current study rather than nuclear-based fecundity differences.

Morphology of fungi was examined under a microscope to assess species. Hyphae from the grapes were collected and DNA extracted using a Chelex based method. The ITS region of fungi was amplified and sequenced using primers ITS1/ITS4 (ITS1, 5′-TCC GTA GGT GAA CCT GCG G-3′; ITS4, 5′-TCC TCC GCT TAT TGA TAT GC-3′). The sequences were compared against GenBank database with BLAST (Basic Logical Alignment Search Tool). Fungal growths were identified as *Talaromyces* and *Penicillium* based on both ITS (Internal Transcribed Spacer) sequencing and morphology. Other microbes, including yeasts, also grow on grapes, but were not identified. 

### 2.3. Comparisons of Lines on Laboratory Medium

We set up male and female pairs from isofemale lines 3 generations after they were established to score life history performance of *w*Au-infected flies on an agar-treacle-dead yeast laboratory medium (c.f. [24]). Pairs of males and females (2 per line, 92 in total) when flies were five days post eclosion were set up on spoons containing the laboratory medium brushed with live yeast paste to encourage egg-laying. Spoons were held at 25 °C and replaced every day over three days (i.e., tested on days 5, 6, and 7 post eclosion). Eggs on spoons were counted to estimate fecundity, and eggs were then transferred to vials with 20 mL of medium to estimate emergence rate, sex ratio, and development time (estimated by scoring emergence daily). Females and males from the same vial were separated into two glass vials and dried at 50 °C for 24 h in an oven. The weight of females and males from each vial was obtained as described above, with vials with less than 4 individuals not weighed.

We compared fecundity, development time, and offspring weight between *w*Au-infected and uninfected lines for each parental age class (5–7 days post eclosion). Data are again presented as violin plots (R software package *vioplot*). Fecundity, emergence rate, and offspring weight (which were normally distributed as tested by Kolmogorov-Smirnov tests) were compared among *w*Au-infected and uninfected lines by ANOVA, with infection and adult parental age as fixed effects. Rank-based tests with the *raov* function from the package, *Rfit*, was used to test the overall effect of infection, adult age, and their interaction on the emergence rate, given that the emergence rate was not normally distributed by a Komogorov-Smirnov test. Development time was compared among *w*Au-infected and uninfected lines with an ANCOVA, which included infection status and adult parental age as fixed effects and emergence number as a covariate because vials, with more larvae having the potential to have a delayed development time due to crowding. Finally, we compared sex ratio between offspring from *w*Au-infected and uninfected lines (pooled across parental age) with Wilcoxon-tests (given that the sex ratio was not normally distributed).

### 2.4. Toxin Effects

*Penicillium* and related fungi growing on grapes produce Ochratoxin A (OTA), which is one of the most-abundant food-contaminating mycotoxins [25]. Eggs from isofemale lines that had been established for 6 generations were tested for toxin responses by rearing larvae in two conditions where the toxin Ochratoxin A (Sigma-Aldrich, St. Louis, MO, USA) had been added to 12 mL of laboratory medium in vials (0.05 ng/mL OTA and 0.5 ng/mL OTA) in medium as well as untreated controls. The lower concentration is relevant to field conditions [26] while the higher concentration was included as an extreme condition. For each line, 30 eggs obtained from spoons were added to medium, and emergence rate, development time, as well as weight of adults were scored.

As in the previous experiment, given that emergence data were not normally distributed, rank-based tests with the *raov* function from the package, *Rfit*, were used to test the overall effect of infection, medium type, and their interaction on the emergence rate. We also compared the emergence rate between *w*Au-infected and uninfected lines of the adults with Wilcoxon-tests and provided kernel density (violin) plots (R software package, *vioplot*). We compared development time (which was normally distributed) between *w*Au-infected and uninfected lines with *t*-tests and used a two-way ANOVA to test for the effects of infection and medium type as fixed effects. Note that because the initial larval density in vials was constant (assuming that the hatchability of the eggs did not vary), we did not include density as a covariate as in the previous experiment. Dry weight was compared between *w*Au-infected and uninfected lines and medium type (fixed effects) with an ANOVA, while *t*-tests were used to compare infection status within medium type.

## 3. Results

### 3.1. Comparison of Lines on Grapes with Fungal Growth

There was an interaction between *Wolbachia* infection and fungus category for the number of offspring produced by 9-day-old parents in the ANOVA (Table 2). Offspring numbers were not significantly different between *w*Au-infected and uninfected flies feeding on low and medium levels of fungus infestations (Figure 1). However, the mean offspring number of *w*Au-infected flies (mean = 39.3, SD = 28.4) was substantially (132%) higher than the mean of the uninfected lines (mean = 16.6, SD = 16.0) when feeding on severely fungus-infested grapes, accounting for the interaction effect (Figure 1). It seems that the *w*Au lines produced larger numbers of offspring than uninfected lines due to higher fecundity and/or larval survival, but only when there was abundant mycelial growth.

For the 13-day-old parents, an interaction of *Wolbachia* infection and fungus was only marginally significant, and the number of offspring produced differed significantly by level of fungus infestation (Table 2). As for the 9-day-old parental comparison, the infected and uninfected parents differed in offspring production; *w*Au-infected flies averaged 43.8 (SD = 34.5) offspring compared to 24.4 (SD = 24.4) offspring for the uninfected flies, reflecting an advantage in offspring production over uninfected lines of 79.5% when grapes were heavily infested (Figure 1). A heavy infestation decreased offspring number.

For the average weight of females and males, the ANOVAs showed a significant difference among fungal levels due to a decrease of 35.8% (females) and 22.7% (males) in weight at higher fungal infestation levels compared to low levels. However, no significant difference was found for female and male weight between *w*Au-infected and uninfected lines (Table 2). The average sex ratio of *w*Au-infected offspring did not differ from that of uninfected offspring (Wilcoxon-test, *p* = 0.637).

### 3.2. Comparison of Lines on Laboratory Medium

Data were obtained from 40 fly pairs from the *w*Au-infected lines and 42 fly pairs from uninfected lines once some replicates without a complete set of parents were removed. Fecundity measured through the number of eggs laid on spoons was significantly different between *w*Au-infected and uninfected lines in 5-day-old parents and 7-day-old parents, with a mean fecundity in infected lines of 37.2 (SD = 7.2) eggs in 5-day-old parents and 27.3 (SD = 12.5) eggs in 7-day-old parents, compared to values of 31.5 (SD = 14.0) and 20.7 (SD = 8.6) for uninfected lines, respectively. For 6-day-old parents, mean fecundities of infected and uninfected pairs differed in the same direction, but this was not significant (Figure 2a). An ANOVA showed no interaction of infection status and parental age. There was a significant effect of infection (Table 3, increase of 20.5% overall in wAu-infected flies), and also for parental age (Table 3).

Sex-ratio was not significantly different in offspring produced by the *w*Au-infected and uninfected flies (Wilcoxon-test, *p* = 0.637). For the emergence rate, the rank-based test (Table 3) also indicated that *w*Au-infected flies produced offspring with similar emergence rates than uninfected flies.

For development time and offspring weight, the ANCOVAs did not show any interaction between the parental age and the infection status, but showed a significant effect of parental age, and a difference between *w*Au-infected and uninfected flies (Table 3). Development time was slowed in offspring from 7-day-old parents by an average of 2% when compared to the offspring of 5-day-old parents, while development time in the *w*Au-infected offspring was also slowed by 1.3% (Figure 2b) compared to the uninfecteds with a mean development time of 10.71 (SD = 0.41) days for the infected offspring compared to 10.60 (SD = 0.40) days for the uninfected offspring. Note that the covariate (emergence number) was significantly (*p* < 0.05) and positively (r = 0.124) associated with development time, reflecting a longer development when larval density was relatively higher. The average weight of female offspring from *w*Au infected offspring was 0.257 (SD = 0.025) mg, which was significantly heavier (Table 3) than the mean weight of uninfected line offspring (mean = 0.251, SD = 0.024), although the increase in weight in *w*Au-infected offspring was relatively small (2.7%). For males, the mean weight of infected offspring was 0.190 (SD = 0.015) mg compared to a mean of 0.186 (SD = 0.0186) mg for uninfected offspring, a significant difference (Table 3) with an increase of 2.5% to also produce extended development times (Figure 2b–d).

### 3.3. Toxin Effects

Emergence rate was lower when the toxin was present and decreased with concentration (Figure 3a), resulting in a significant effect of the OTA condition on this variable (Table 4). There was no interaction between infection and the OTA condition (Table 4), but a marginally significant overall effect of the infection due to the somewhat higher emergence of the infected lines (89.4%) compared to the control lines (86.9%).

For development time, the ANOVA (Table 4) showed a significant difference between *w*Au-infected and uninfected individuals (Table 4). The *w*Au-infected larvae had a longer development time, averaging 11.04 (SD = 0.098) days compared to the uninfecteds, which averaged 10.94 (SD = 0.13) days, representing an increase of 0.86% for the infected larvae. This increase was particularly evident for normal medium and for medium with 0.05ng/mL OTA added (Figure 3b).

The weight of female adults infected with *w*Au was higher than for uninfected females (Figure 3c,d) regardless of whether there was OTA in the medium; the effects of the infection and toxin were significant in the ANOVA and there was also an interaction effect (Table 4). On average, the female weight of infected flies (mean = 0.362, SD = 0.029 mg) was higher than that of the uninfected flies (mean = 0.336, SD = 0.040 mg) with an overall difference of 0.076%. The toxin presence decreased female weight, particularly at the highest concentration tested (Figure 3). There was no impact of the toxin on male weight, although there was a marginally significant effect of the infection on male weight (Table 4).

## 4. Discussion

The current work points to fitness advantages as well as costs associated with the *w*Au infection of *D. simulans*. The fitness advantages relate to a larger size (particularly in females) and a higher fecundity, likely to be a reflection of this larger size, while the cost is associated with an extended development time, again likely reflecting the fact that *w*Au-infected individuals would have pupated at a larger larval size (although this was not measured directly). These fitness effects probably contributed to the advantage that *w*Au-infected hosts showed under conditions where flies were reared on fruit contaminated with mycelial growth where a marked difference in productivity was detected. In particular, we anticipated that the higher fecundity of the infected lines observed on laboratory medium increased productivity although it is not obvious why this was only evident when there was mycelial growth. Perhaps there is an additional advantage stemming from a slower development time when mycelial growth is present (or there are other concomitant changes in microbes including yeasts), which allows the larvae to develop under the stressful fungal conditions (stress being evident from the small size of the emerging flies and lower emergence rate). Perhaps infected individuals somehow promote the growth of fungi. Although we were unable to link effects to one of the chemicals produced by the fungi (OTA), other compounds associated with microbial growth might still be important. To further investigate the potential effects of microbes on *Wolbachia* fitness, additional experiments are required, such as culturing specific strains of fungi, yeasts, and so on under sterile conditions, where nutrition can be carefully controlled [27].

Previous results from Kriesner et al. [20] highlighted the strong fitness effect of *w*Au in population cages, where flies were held with laboratory medium at a very large population size. These authors included two experimental treatments, where medium was “conditioned” to increase potential microbial activity by being exposed to fly populations to allow for oral or faecal transmission, as well as a situation where fruit was not conditioned before being used as a breeding site. However, these treatments did not influence the relative advantage of the *w*Au infection in *D. simulans* as reflected by the rate of increase of the infection in population cages. Instead, we suspect that based on the current results, there was an advantage of *w*Au associated with some of the fitness effects detected in the current study, although the advantage of *w*Au-infected hosts under conditions of abundant fungal mycelial growth would not have been a component of that study.

Our findings contrast with some previous work to identify fitness differences associated with the *w*Au-infected strain. In the initial paper on this strain, Hoffmann et al. [18] suggested that *w*Au acted as a neutral variant in natural *D. simulans* populations, being transmitted maternally at a high incidence near 100%, but not having any notable deleterious fitness effects (and not causing CI). The fitness effect measured in that study was fecundity, and it was measured by comparing an infected population with a derived population cured by exposure to tetracycline for a generation. However, the “cured” line was only given one generation without tetracycline to recover whereas it is now known that recovery from antibiotic treatment can take several generations [28], although the life history effect of tetracycline treatment unrelated to *Wolbachia* is unknown. The apparent absence of *w*Au fitness effects is also contradicted by the rapid increases in *w*Au frequency seen in natural populations of *D. simulans* in Australia prior to the arrival of the *w*Ri infection, which has subsequently spread through all eastern Australian populations due to CI [24]. The relatively high frequency of *w*Au we observed here in a Western Australian population also confirms previous observations [24] that *w*Au has increased from a low frequency in Australia where it was first detected [18] to being relatively common in populations.

In the current study, a different experimental design was used to measure fitness effects associated with *Wolbachia* than is normally applied in fitness comparisons. We did not use tetracycline curing, which has been commonly applied, with the advantage that this approach controls for differences in nuclear background at least in the absence of genetic drift in lines after curing. In our design, the comparison of infected and uninfected lines involved multiple independently-derived isofemale lines obtained from the same field sample. This means that the nuclear genetic background of the infected and uninfected lines should both be representative of what is found in the field (given that each isofemale line captures a substantial fraction of genetic variation from the field population [29]), and randomized with respect to the infection given that nuclear-*Wolbachia* disequilibrium is not expected in populations [30]. While we averaged the effects of the infection across a sample of 20 or more nuclear backgrounds maintained identically as uninfected isofemale lines, fitness effects associated with *Wolbachia* infections are often undertaken through a comparison of one “cured” and uncured line or a comparison of one naturally infected or uninfected population. Comparisons based on a limited set of lines can include repeated backcrossing to homogenize lines to a common stock or reciprocal crosses to increase the likelihood that any detected effects are linked to *Wolbachia*, but the current design has the added advantage that inferences of infection effects are obtained on randomized nuclear backgrounds that are representative of those in the sampled field population. Obviously, such a design is much simpler to achieve in *Drosophila*, where a large number of lines can be readily reared as compared to in (say) mosquitoes, where line culture across multiple populations is much more challenging.

Within the limitations of generating meaningful fitness estimates, other studies have pointed to life history advantages of *Wolbachia*. In particular, the *w*Ri infection of *D. simulans* was originally considered deleterious, but has evolved to increase fecundity [31]. The *Wolbachia* infection in *D*. *suzukii* is also expected to be beneficial given that it has increased in frequency, but does not cause CI [17] and there is evidence for a fitness advantage for *Wolbachia* associated with *D. mauritiana* [32]. Fitness benefits of *Wolbachia* and other endosymbionts have been well established in other organisms, particularly through the provision of nutrition. These include beneficial effects of *Cardinium* and *Wolbachia* on fecundity and survival of spider mites [33] and of *Wolbachia* on the growth and reproduction of bed bugs [14]. In some cases, the basis for fitness benefits have been established, such as the provisioning of riboflavin by *Wolbachia* in bedbugs [14] and the provisioning of amino acids by *Buchnera* in aphids [2,34]. However, there is also a level of inconsistency in reported fitness effects associated with *Wolbachia* infections, particularly in mosquitoes [35].

Apart from showing that life history traits are altered by the *w*Au infection, our experiments also tested for environment-specific fitness effects following the results linked to mycelial growth. In this respect, the toxin experiment failed to show specific impacts under certain conditions. In discussing fitness effects associated with *Wolbachia*, there is often a tendency in the literature to immediately attribute fitness associations to environment-specific components. In particular, the reduction in viral transmission associated with many *Wolbachia* strains in *Drosophila* [36,37] is seen as a way in which *Wolbachia* infections can be favoured in conditions where virus is present. However, at least in *Drosophila*, it is not yet apparent whether this applies; although several *Wolbachia* strains block viruses under laboratory conditions, natural viruses from field-collected flies and lines have not yet been linked to the presence of *Wolbachia*, at least with respect to the *Wolbachia* infection from *D. melanogaster* [38] that is persistently polymorphic in natural populations [19,39].

Our results also point to effects of parental age on the life history traits. This includes a reduction in offspring number when flies were reared on grapes, and a decrease in fecundity, increase in development time, and inconsistent effect of offspring weight in older parents. Female age effects on reproduction have been reported previously (e.g., [40]), and in our case, may reflect an exhaustion of eggs in older females, while the longer development time and weight changes may reflect changes in provisioning of eggs in older females. However, apparent age effects should be treated cautiously because the flies were transferred to new media/fruit, which may have differed from the initial batch used.

## 5. Conclusions

In conclusion, we found life history effects associated with the *w*Au infection that provide an advantage and potential cost to flies under at least one set of laboratory conditions, expressed under a field-relevant nuclear background. The fitness effects may be partly mediated through changes in fly size, a trait which has previously been related to the *w*Mel infection in some (but not all) field collections [41]. Knowledge of such fitness effects is important in predicting the ability of infections to invade natural populations [22], particularly in the absence of cytoplasmic incompatibility. The advantage of *w*Au over uninfected flies may account for the increase in the frequency of this infection both in population cages and under field conditions seen in Australian populations [20,24]. The generality of these findings remains to be tested with other infections and hosts, but are likely to differ depending on *Wolbachia* density, tropism, and other factors [35]. The genomic basis of these effects also remains to be determined; the wAu infection lacks a *w*Mel ortholog involved in cytoplasmic incompatibility, but also has numerous other differences [42] that may be linked to life history effects.

## Figures and Tables

**Figure 1 insects-10-00126-f001:**
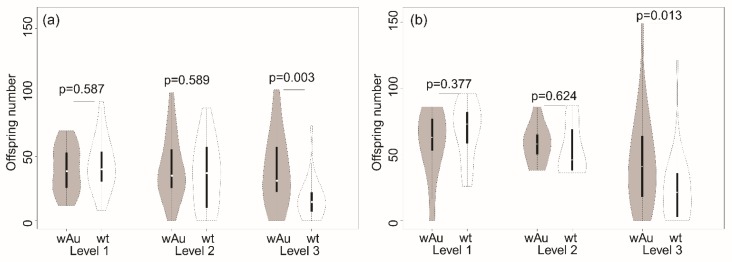
Violin plots for distribution of offspring number of *w*Au-infected and uninfected (wt) lines at different levels of fungal growth. (**a**) Offspring number of 9-day-old parents; (**b**) offspring number of 13-day-old parents. Levels 1 to 3 indicate the levels of fungus infestation on grapes as described in the text. Medians are indicated by central points, and the interquartile range by the solid bar, while the dotted line indicates the 95% confidence interval.

**Figure 2 insects-10-00126-f002:**
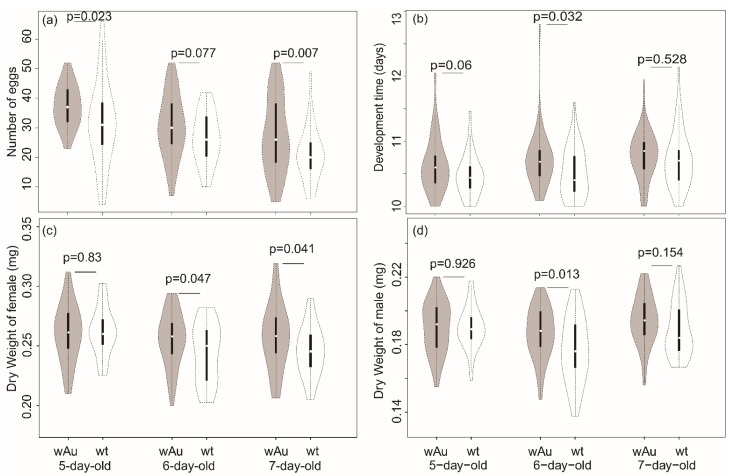
Violin plots for (**a**) fecundity (number of eggs), (**b**) development time, (**c**) female weight, and (**d**) male weight of *w*Au-infected and uninfected lines reared on laboratory medium with parents transferred across three days to new medium (5-, 6-, and 7-day-old parents). Medians are indicated by central points, and the interquartile range by the solid bar, while the dotted line indicates the 95% confidence interval.

**Figure 3 insects-10-00126-f003:**
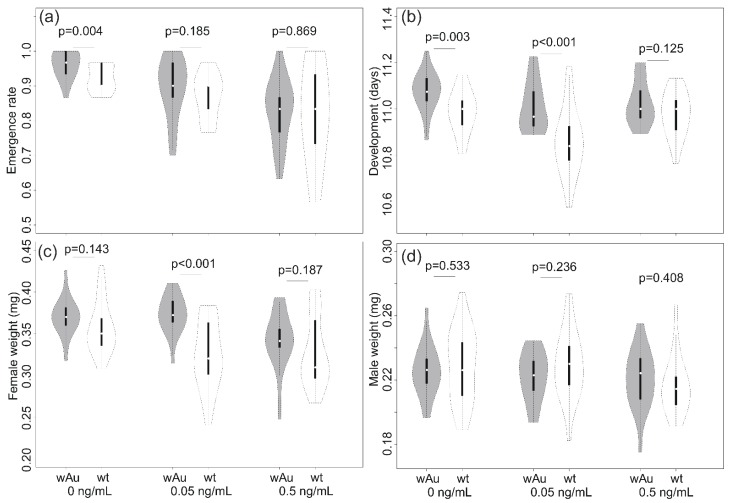
Violin plots for (**a**) emergence rate, (**b**) development time, (**c**) female weight, and (**d**) male weight of *w*Au-infected and uninfected (wt) larvae reared on three medium conditions (no OTA (Ochratoxin), 0.05 ng/mL OTA, and 0.5 ng/mL OTA in medium). Medians are indicated by central points, and the interquartile range by the solid bar, while the dotted line indicates the 95% confidence interval.

**Table 1 insects-10-00126-t001:** Outline of experiments undertaken.

Experiment	Number of Isofemale Lines Used	Generation Since Isofemale Lines Established	Traits Scored
Comparison of lines on grapes with fungal growth	23 *w*Au-infected and 23 uninfected lines	2	Number of offspring produced, dry weight, sex ratio
Comparisons of lines on laboratory medium	As above	3	Fecundity, egg-to-adult emergence, sex ratio, development time, dry weight
Toxin effects	As above	6	Egg-to-adult emergence, development time, dry weight

**Table 2 insects-10-00126-t002:** ANOVAs (MS = mean square, F ratio, probability) for offspring number and offspring weight when flies were allowed to produce offspring on grapes with fungal growth; parental flies produced offspring in one set of bottles when they were 9 to 11 days post-eclosion, and a second set of bottles when they were 12 to 14 days post-eclosion. Significant results (*p* < 0.05) are indicated in bold.

Variable	Infection (df = 1)		Fungal Level (df = 2)		Infection by Fungal Level (df = 2)		Error MS (df)
	MS	F (P)	MS	F (P)	MS	F (P)	
Offspring number (parents 9–11 days old)	2431	**4.708 (0.032)**	2355	**4.561 (0.012)**	1875	**3.631 (0.030)**	516 (118)
Offspring number (parents 12–14 days old)	2519	3.579 (0.061)	10921	**15.52 (<0.001)**	1884	2.676 (0.074)	704 (104)
Weight (mg, female)	0.0000	0.001 (0.974)	0.1577	**37.99 (<0.001)**	0.0012	0.292 (0.747)	0.0042 (191)
Weight (mg, male)	0.0008	0.687 (0.408)	0.0259	**21.12 (<0.001)**	0.0005	0.386 (0.680)	0.0012 (193)

**Table 3 insects-10-00126-t003:** ANOVAs (MS = mean square or MRD = mean square for emergence based on ranked data, F ratio, probability) for fecundity and emergence rate, as well as an ANCOVA performed on preimaginal development when flies were reared on laboratory medium and transferred daily for three days to produce different parental ages. Significant results (*p* < 0.05) are indicated in bold. The ANCOVA on development time included the emergence number as a covariate.

Variable	Infection (df = 1)		Age (df = 2)		Infection by Age (df = 2)		Error MS (df)
	MS/MRD	F (P)	MS/MRD	F (P)	MS/MRD	F (P)	
Fecundity	1778.9	**15.96 (<0.001)**	2206.1	**19.80 (<0.001)**	39.0	0.350 (0.705)	111.4 (240)
Emergence rate (ranked data)	0.00015	0.0012(0.972)	0.0529	0.441 (0.644)	0.0495	0.413 (0.662)	- (234)
Development time in days ^1^	1.1435	**7.393 (0.007)**	0.8082	**5.225 (0.007)**	0.1354	0.875 (0.418)	0.1585 (229)
Weight (mg, female)	0.0026	**4.532 (0.034)**	0.0031	**5.524 (0.005)**	0.0011	1.929 (0.148)	0.5530 (208)
Weight (mg, male)	0.0011	**4.103 (0.044)**	0.0015	**5.693 (0.004)**	0.0003	1.094 (0.337)	0.0003 (196)

^1^ ANCOVA with the emergence number included in the analysis. The number of flies emerged had a significant effect on development time (F_(1,229)_ = 6.373, *p* = 0.012).

**Table 4 insects-10-00126-t004:** ANOVAs (MS = mean square or MRD = mean square for emergence based on ranked data, F ratio, probability) for emergence rate, adult weight, and preimaginal development when flies were reared on medium with different OTA (Ochratoxin) levels. Significant results (*p* < 0.05) are indicated in bold.

Variable	Infection (df = 1)		OTA (df = 2)		Infection by OTA (df = 2)		Error MS (df)
	MS/MRD	F (P)	MS/MRD	F (P)	MS/MRD	F (P)	
Emergence rate (ranked data)	0.1289	**4.733 (0.032)**	0.6885	**25.277 (<0.001)**	0.0076	0.280 (0.756)	- (120)
Development time in days	0.2801	**25.419 (<0.001)**	0.1027	**9.318 (<0.001)**	0.0289	2.619 (0.077)	0.0110 (120)
Weight (mg, female)	0.021	**19.4 (<0.001)**	0.009	**7.999 (<0.001)**	0.004	**4.042 (0.020)**	0.0011 (120)
Weight (mg, male)	0.000	0.359 (0.550)	0.001	2.846 (0.062)	0.000	1.075 (0.344)	0.0003 (120)

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
