# Peer review of "Life History Effects Linked to an Advantage for *w*Au *Wolbachia* in *Drosophila"

_insects, 2019, doi:10.3390/insects10050126_

Round 1

Reviewer 1 Report

The paper entitled « Life history effects linked to an advantage for wAu Wolbachia in Drosophila » by Cao et al. focuses on the association between Drosophila simulans and the strain of the bacterial symbiont Wolbachia wAu. This strain does not induce any manipulation of the reproduction in its native host but is still maintained in field populations. The present study aims at testing fitness effects of this Wolbachia infection on several life history traits and different medium in D. simulans. The authors show that wAu can provide fitness benefits in presence of fungi on grapes, but also on laboratory medium. This work has the originality to have been done on many isofemale lines (infected by wAu and uninfected) obtained from individuals collected in the field that have thus a diversity of nuclear backgrounds that are representative of those in the sampled field population, instead of trying to obtain infected and uninfected lines with a homogeneous nuclear background obtained using antibiotics treatments to cure Wolbachia or introgressions. This results in a huge amount of work that certainly merit to be published but requires to be improved. Indeed, some parts are very difficult to follow (“material and methods”, “results”) and should be reorganized, clarified and simplified. The introduction is very short and the examples are mainly based on Wolbachia and especially Wolbachia in Drosophila while other bacterial endosymbionts are known to provide direct fitness benefits, and other model insects such as aphids have been the purpose of many studies. In the same way, the discussion is short and there are few comparisons with other studies (other than wAu in D. simulans). Moreover, I have some problems with statistics: when ANOVA2 show an interaction between the two factors, it is not possible to conclude on the separate effect of the two factors.

Please find below other comments:

Introduction:

-          As I already said, the examples should be more diverse with studies that focused on other bacterial symbionts and other insects.

-          L. 48-52: the phenotypic effect of bacterial endosymbionts is known to be genotype-dependent, therefore, it is not because wAu infection is associated with positive effects in D. simulans that it would also provide direct fitness benefits to another host!

-          L. 55-57: it is not explained why fungi have been chosen here. MS [15] and [16] that are cited do not mention them.

Material and methods:

-          Globally this part should seriously be reorganized. There is redundancy between experiments: for example, the weight has been determined in the same way for the three experiments… having a paragraph with the general protocol for the different LHT measures and another with the statistical analyses would allow shortening this part. When there are some differences they can be added elsewhere, in the methods part or in the results.

-          For helping the reader, a scheme of the experiments would be welcome, with the three parts indicated.

-          There is a lack of details concerning the lines’ establishment and the individuals used for the different experiments, this should be clarified. For example, L. 65/66, how many offspring per mother has been use to produce the isofemale lines?

-          L. 69: 60% of Wolbachia infection: is there a reference or does it come from the present study?

-          L. 73/75: the HRM technic would have deserved to be a little bit explained. What is the general principle? Moreover, primers have been given for Drosophila, why not the primers for Wolbachia?

-          L. 77: this also allow testing for the quality of the extract not only for the presence of the DNA.

-          L. 78/82: there is no wRi control for the HRM???

-          L. 90: 3 pairs of uninfected and wAu-infected males-females/isofemale line, how can you have 23 pairs in total?

-          L. 90/103: again a scheme would have been useful.

-          Kernel density (violin) plots is not an usual representation and it would merit to be explained at least once (may be in the “materials” part). Moreover, why having chosen this representation? What is the added value compared to “classical” box plots?

-          L. 136/141: Why having chosen to focus on the search of fungi and not also added primers for detecting possible bacteria?

-          It’s a shame that there is no measure of Wolbachia density under the different conditions and especially in presence/absence of the fungi or OTA because if the density of Wolbachia changes it can affect the rate of its transmission.

-          L. 146: the 23 uninfected and infected pairs of males and females correspond to what we could call G0, i. e. individuals used in the first experiments on grapes? And then G0 and G1 have been developed on grapes and the F2 used here on lab medium? L. 152: the weight has been measured on F2 individuals? Please clarify. If it is the case, did the individuals used were developed on grapes with or without fungi? And what about maternal and trans-generational effects? This should be discussed!

-          Is the weight determined for the whole vial and then, based on the number of individuals, the average weight of all individuals has been determined (so all individuals of the same vial are considered to have on average the same weight)? Is it that? If yes, why didn’t you choose to measure the fresh weight which would allow having individual measures?

-          For the experiment with the toxin, do the eggs from the F5 come from the same G0 lines (5 generations after)?

-          How did you choose the toxin’s concentrations? May be a higher concentration would allow seeing more effects?

Results:

-          L. 192-200 (and again l. 275): there is an interaction between the 2 factors so you can’t conclude about the effect of each factor separately.

-          Table 1: MS for females’ weight: how is it possible to obtain 0?

-          Table 1: what does the column “Error MS” mean? Please indicate it clearly.

-          L. 208: SD are huge! How is it possible to have a SD equivalent to the mean (24.4), is it an error?

Discussion/conclusion:

-          L. 290: if an extended development time really reflects a larger larval size, is it really a cost?

-          L. 291: OK but how can you explain that a selective advantage has also been observed on the laboratory medium?

-          The influence of the age of the parents has not been discussed while for some (most) traits it has a significant effect.

-          More generally results are not enough discussed. For example, what are the consequences of fitness advantage bring by wAu on the competition with other species? What can be the consequences on the competition with D. simulans infecting with other Wolbachia strain and thus on wAu prevalence in field populations?

-          L. 298/299: other concomitant changes of microbes have just been evoked but would have merit to be more discussed (bacteria, yeast).

-          The two first paragraph of page 9 should be more developed. In the second paragraph, the first part is about other protocols used to have infected and uninfected lines of D. simulans based on antibiotic treatments. I agree that these treatments can have an impact on several generations. However, breeding on grapes with fungi could also impact offspring (trans-generational effects).

-          The third paragraph of the page 9 explains that the protocol used here allows using representative nuclear backgrounds and host-wAu interactions in the sampled field. I agree but it is a huge work, with a lot of lines to manage compared with the introgressions’ methods!

-          L. 345-352: one more time, the discussion is very short and does not replace the results obtained with other results and mainly focuses on Wolbachia-Drosophila interactions.

Other details:

-          L. 25: change the last character “,” by “.”

-          L. 47: add “s” at “fruit”

-          L. 90: add a “s” at “pair”

-          L. 113 and 163: “l” is lacking at “Kolmogorov”

-          Figure 1: indicate that “wt” means uninfected

-          L. 131: delete “but”

-          L. 133 to 135: should be rephrased

-          L. 149: “i. e.” should be in italics

-          L. 150: add a “n” at “then transferred”

-          L. 159: add “as tested” before “by”

-          L. 192: Rephrase “differed by ANOVA”

-          L. 202: Legend of Table 1: “transferred for three days”, please complete

-          L. 205: “between” level rather than “by”

-          L. 210: “under this treatment”, please clarify

-          L. 218: delete “as”

-          L. 226: replace “although” by “and”

Moreover, in ANOVA2, the first hypothesis to be tested should be whether there is interaction between the two factors and if no, the other two hypotheses can be tested.

-          Table 2, L. 231: please add MS before “Mean Square” as in table 3, L. 233: please clarify the transfer, L. 234: the ANCOVA “performed” on development time

-          L. 236/237: reformulate. If I understand well: “The number of emergences was included in the ANCOVA and had a significant effect on development time

-          L. 238: add (and rewrite the ANCOVAs “didn’t show any interaction between the parental age and the infection status but showed a significant effect of the two factors separately”

-          L. 243: delete “)” before “with”

-          L. 246: delete “separately”

-          L. 247: add “)” after “mg”

-          L. 251: replace “and” increase by “with an” increase; “to also produce extended”, please rephrase

-          L. 259-262: as said previously, the first hypothesis to be tested should be whether there is interaction between the two factors

-          L. 268/268: please delete “and a marginally non-significant interaction with OTA condition”

-          L. 294: what does “productivity eventuated” means?

-          L. 294/295: reformulate “helped contribute greater productivity”

-          L. 333: add “)” after [33]

-          L. 346: rephrase “suggestive evidence”

-          L. 358: delete “evident”

Author Response

The paper entitled « Life history effects linked to an advantage for wAu Wolbachia in Drosophila » by Cao et al. focuses on the association between Drosophila simulans and the strain of the bacterial symbiont Wolbachia wAu. This strain does not induce any manipulation of the reproduction in its native host but is still maintained in field populations. The present study aims at testing fitness effects of this Wolbachia infection on several life history traits and different medium in D. simulans. The authors show that wAu can provide fitness benefits in presence of fungi on grapes, but also on laboratory medium. This work has the originality to have been done on many isofemale lines (infected by wAu and uninfected) obtained from individuals collected in the field that have thus a diversity of nuclear backgrounds that are representative of those in the sampled field population, instead of trying to obtain infected and uninfected lines with a homogeneous nuclear background obtained using antibiotics treatments to cure Wolbachia or introgressions. This results in a huge amount of work that certainly merit to be published but requires to be improved. Indeed, some parts are very difficult to follow (“material and methods”, “results”) and should be reorganized, clarified and simplified. The introduction is very short and the examples are mainly based on Wolbachia and especially Wolbachia in Drosophila while other bacterial endosymbionts are known to provide direct fitness benefits, and other model insects such as aphids have been the purpose of many studies. In the same way, the discussion is short and there are few comparisons with other studies (other than wAu in D. simulans). Moreover, I have some problems with statistics: when ANOVA2 show an interaction between the two factors, it is not possible to conclude on the separate effect of the two factors.

Thanks for your detailed review.  Please see responses below. We have covered these general points in the changes made in response to the detailed comments, including broadening the reference to other work, altering the methods, and changing the statistical presentation.

Please find below other comments:

Introduction:

-          As I already said, the examples should be more diverse with studies that focused on other bacterial symbionts and other insects.

Sure, we’ve now added a paragraph up front about the diverse effects of endosymbionts on other insects to provide a more general context. Lines 29-37.

-          L. 48-52: the phenotypic effect of bacterial endosymbionts is known to be genotype-dependent, therefore, it is not because wAu infection is associated with positive effects in D. simulans that it would also provide direct fitness benefits to another host!

We appreciate this point but then again very little work has been done on transinfections where there are host benefits. However we now have added a short statement about the native host (line 64). Also note that we emphasized interest in this infection in mosquitoes due to virus blocking effects.

-          L. 55-57: it is not explained why fungi have been chosen here. MS [15] and [16] that are cited do not mention them.

 MS 19 and 20 point to the fitness advantage of the infection which is what the statement indicates (“which pointed to fitness advantages of this infection”). We chose fungi because fungal growth is a rather obvious feature of rotting fruit in Drosophila breeding sites but we have made a few minor wording changes to clarify this.

Material and methods:

-          Globally this part should seriously be reorganized. There is redundancy between experiments: for example, the weight has been determined in the same way for the three experiments… having a paragraph with the general protocol for the different LHT measures and another with the statistical analyses would allow shortening this part. When there are some differences they can be added elsewhere, in the methods part or in the results.

We’ve done reorganization around weight where general methods were applicable and also cut down on wording around other traits scored in the same way. However the experiments (which are now tabulated – see Table 1) are quite different and the traits/analysis also differs so we prefer to keep this separate as in the original version (we did look at changing this but it became quite messy!).

-          For helping the reader, a scheme of the experiments would be welcome, with the three parts indicated.

A table is now provided outlining the experiments and some details about them. See Table 1.

-          There is a lack of details concerning the lines’ establishment and the individuals used for the different experiments, this should be clarified. For example, L. 65/66, how many offspring per mother has been use to produce the isofemale lines?

This is indicated on line 77. Some details have been moved to Table 1 to assist a reader.

-          L. 69: 60% of Wolbachia infection: is there a reference or does it come from the present study?

Current study. Now mentioned, line 81.

-          L. 73/75: the HRM technic would have deserved to be a little bit explained. What is the general principle? Moreover, primers have been given for Drosophila, why not the primers for Wolbachia?

The HRM approach has been reported in several papers from our lab so we don’t want to repeat this. It is taken from Lee et al as cited. However we have now provided some additional details including a specific reference to the primers. Lines 87-90.

-          L. 77: this also allow testing for the quality of the extract not only for the presence of the DNA.

Now mentioned. See line 94.

-          L. 78/82: there is no wRi control for the HRM???

There is and it was used as in Kriesner et al, now added.

-          L. 90: 3 pairs of uninfected and wAu-infected males-females/isofemale line, how can you have 23 pairs in total?

The 3 refers to replicates per line, now reworded. Lines 110-1.

-          L. 90/103: again a scheme would have been useful.

Hopefully the table helps.

-          Kernel density (violin) plots is not an usual representation and it would merit to be explained at least once (may be in the “materials” part). Moreover, why having chosen this representation? What is the added value compared to “classical” box plots?

The advantage of these is that they provide a more complete picture of the distribution of points than box plots (for instance the tails are much clearer) as well as providing similar information to box plots. They are being increasingly used in the literature. We have added an extra statement on them and described them further in the legends. Line 137 and legends.

-          L. 136/141: Why having chosen to focus on the search of fungi and not also added primers for detecting possible bacteria?

The fungal growth is rather obvious in these experiments, and we were not in a position to test all microbes including yeasts quantitatively.  

-          It’s a shame that there is no measure of Wolbachia density under the different conditions and especially in presence/absence of the fungi or OTA because if the density of Wolbachia changes it can affect the rate of its transmission.

This information is not available and beyond the scope of the current work.

-          L. 146: the 23 uninfected and infected pairs of males and females correspond to what we could call G0, i. e. individuals used in the first experiments on grapes? And then G0 and G1 have been developed on grapes and the F2 used here on lab medium? L. 152: the weight has been measured on F2 individuals? Please clarify. If it is the case, did the individuals used were developed on grapes with or without fungi? And what about maternal and trans-generational effects? This should be discussed!

There seems to be some confusion here. All lines when set up were reared on laboratory medium for two generations when establishing the lines. This is in the methods (see line 86). A line is established with offspring of a female, and the first experiment was undertaken with lines established for 2 generations. This information is added to Table 1. Flies for experiments were always reared on lab medium (offspring from the grape experiment were not used to continue the lines). Thus there is no transgenerational environmental effect. See line 82. We realize some confusion may have been caused by the use of F2 etc, which refers to generation of lab lines since establishment and this has now been altered.

-          Is the weight determined for the whole vial and then, based on the number of individuals, the average weight of all individuals has been determined (so all individuals of the same vial are considered to have on average the same weight)? Is it that? If yes, why didn’t you choose to measure the fresh weight which would allow having individual measures?

Yes that is correct, as now further noted. The problem with fresh weight is that flies are collected across a few days and fresh weight can change with age, hence our focus on dry weight.

-          For the experiment with the toxin, do the eggs from the F5 come from the same G0 lines (5 generations after)?

Yes the same lines, but with 6 generations elapsed since lines were established (F5 plus parents), this is now rephrased in the MS and see Table 1.

-          How did you choose the toxin’s concentrations? May be a higher concentration would allow seeing more effects?

The amount of OTA (0.05 ng /mL) was determined according to Mutlu (2012), which is the amount relevant to natural conditions. A higher concentration (0.5 ng OTA/mL) was also tested to see if there were any effects. This is now mentioned at lines 195-6.

Mutlu, A. G. Increase in mitochondrial DNA copy number in response to Ochratoxin A and methanol-induced mitochondrial DNA damage in Drosophila. Bull Environ Contam Toxicol 2012, 89, 1129–1132.

Results:

-          L. 192-200 (and again l. 275): there is an interaction between the 2 factors so you can’t conclude about the effect of each factor separately.

Changed.

-          Table 1: MS for females’ weight: how is it possible to obtain 0?

If there is a very small SS due to this term, it rounds down to 0.

-          Table 1: what does the column “Error MS” mean? Please indicate it clearly.

Not sure why there is confusion around this, it is the MS due to error in the ANOVA. MS now explained in legend.

-          L. 208: SD are huge! How is it possible to have a SD equivalent to the mean (24.4), is it an error?

Note SD, not SE. Unfortunately there are some outlier scores, as evident in the plots. We did redo the analyses after removing these/transforming the data but it did not affect the conclusions.

Discussion/conclusion:

-          L. 290: if an extended development time really reflects a larger larval size, is it really a cost?

Extended development time is certainly a fitness cost which may be partly/completely compensated by size, but we don’t know that. These traits can be independent in Drosophila such as along a latitudinal cline so we have not changed this.

-          L. 291: OK but how can you explain that a selective advantage has also been observed on the laboratory medium?

The large size was seen regardless of medium, and we now note that the fecundity increase likely associated with this size difference may help contribute to the offspring number difference on fruit. See lines 316-8.

-          The influence of the age of the parents has not been discussed while for some (most) traits it has a significant effect.

We’ve now mentioned more on this in a specific paragraph (lines 396-403), though it is always hard to interpret such trends given that flies were transferred to new media at later ages which is a confounding effect.

-          More generally results are not enough discussed. For example, what are the consequences of fitness advantage bring by wAu on the competition with other species? What can be the consequences on the competition with D. simulans infecting with other Wolbachia strain and thus on wAu prevalence in field populations?

More on this has been added at line 410-412. It may explain the increase in wAu observed previously although there are developmental costs. We think it is premature to speculate on species competition effects.

-          L. 298/299: other concomitant changes of microbes have just been evoked but would have merit to be more discussed (bacteria, yeast).

We don’t want to speculate too much on this (it would be pure speculation) but some mention has been made and this really requires more experimental work (see lines 322-327).

-          The two first paragraph of page 9 should be more developed. In the second paragraph, the first part is about other protocols used to have infected and uninfected lines of D. simulans based on antibiotic treatments. I agree that these treatments can have an impact on several generations. However, breeding on grapes with fungi could also impact offspring (trans-generational effects).

But we are not using these offspring in the ensuing experiments. This is now emphasized in the methods (see above). All flies used in subsequent experiments came from the same isofemale lines only reared under laboratory conditions.

-          The third paragraph of the page 9 explains that the protocol used here allows using representative nuclear backgrounds and host-wAu interactions in the sampled field. I agree but it is a huge work, with a lot of lines to manage compared with the introgressions’ methods!

The introgression approach has the issue that the background used for introgression may be atypical. Yes it is a lot of work but at least in Drosophila and some other model species it is achievable, which is a core message in this paper.

-          L. 345-352: one more time, the discussion is very short and does not replace the results obtained with other results and mainly focuses on Wolbachia-Drosophilainteractions.

We’ve expanded this somewhat and mentioned other systems as per the introduction. See lines 377-383.

Other details:

-          L. 25: change the last character “,” by “.”. Corrected

-          L. 47: add “s” at “fruit”. Changed

-          L. 90: add a “s” at “pair”. Corrected

-          L. 113 and 163: “l” is lacking at “Kolmogorov”. Corrected

-          Figure 1: indicate that “wt” means uninfected. Added.

-          L. 131: delete “but”. Changed.

-          L. 133 to 135: should be rephrased

-          L. 149: “i. e.” should be in italics. Changed

-          L. 150: add a “n” at “thetransferred”. Corrected

-          L. 159: add “as tested” before “by”. Changed

-          L. 192: Rephrase “differed by ANOVA”. Changed

-          L. 202: Legend of Table 1: “transferred for three days”, please complete. We’ve clarified this by referring here to parental age, also see Table variable names.

-          L. 205: “between” level rather than “by”. Changed

-          L. 210: “under this treatment”, please clarify. Now clarified

-          L. 218: delete “as”. Changed

-          L. 226: replace “although” by “and”. Changed

Moreover, in ANOVA2, the first hypothesis to be tested should be whether there is interaction between the two factors and if no, the other two hypotheses can be tested. OK swapped.

-          Table 2, L. 231: please add MS before “Mean Square” as in table 3, L. 233: please clarify the transfer, L. 234: the ANCOVA “performed” on development time. Clarifications made

-          L. 236/237: reformulate. If I understand well: “The number of emergences was included in the ANCOVA and had a significant effect on development time. OK changed.

-          L. 238: add (and rewrite the ANCOVAs “didn’t show any interaction between the parental age and the infection status but showed a significant effect of the two factors separately”. Changed.

-          L. 243: delete “)” before “with”. Done

-          L. 246: delete “separately”. Not at 246

-          L. 247: add “)” after “mg”. Done

-          L. 251: replace “and” increase by “with an” increase; “to also produce extended”, please rephrase. Done

-          L. 259-262: as said previously, the first hypothesis to be tested should be whether there is interaction between the two factors. OK swapped

-          L. 268/268: please delete “and a marginally non-significant interaction with OTA condition”. Done

-          L. 294: what does “productivity eventuated” means? Changed

-          L. 294/295: reformulate “helped contribute greater productivity”. Done

-          L. 333: add “)” after [33]. Done

-          L. 346: rephrase “suggestive evidence”. Done

-          L. 358: delete “evident”. Done

Reviewer 2 Report

Wolbachia infections are common in insects, and the questions as to how they persist and spread could be important in both basic and applied science. While Wolbachia infections with little influence on host reproduction (e.g., CI, feminization, or male killing, etc.) can still spread as found in earlier studies from this same group, underlying mechanisms are still unclear in some situations. In the current study, the authors show wAu infected flies have relative fitness advantage (i.e., higher offspring number) when fungi are growing heavily whereas such effect is not pronounced when fungi are less in breeding sites. The observation is quite interesting, and this could be first interpreted that the infection has a fitness advantage when the environment is harsh, which could potentially explain the wide spread of wAu in natural fly populations.  Although the authors could not replicate the results in another experiment that looked at the effect of wAu infection on the life history traits of flies in toxin (OTA) contaminated breeding sites, it might also suggest that there could be other factors that mediated such effect as the effect of fungi (and other things growing on the grapes) could be quite complex.  Perhaps, wAu infection may benefit from some substances produced by fungi or something else. The manuscript is well written overall. The experiments are carefully designed and performed, and the data are analyzed properly with clear presentation.  The application of fluctuating temperature as a norm for baseline experimental environment is notable.  I have no further suggestions except for the minor comments below.

1.      Experimental design of fungus exposure – I wonder if it would be possible to control the amount of fungi by inoculating known amount.  Looking at Figure 1, could fungi grow more when there are more infected flies?

2.      Line 136 – Perhaps authors could add a bit more details as to how growing of wanted species of fungus could be controlled. Are there any complication in interpreting the results linking to the previous study and the OTA experiment in the current study considering many other things could potentially grow in grapes?

3.      Line 174 – How the amount of OTA was determined? Is the amount relevant to natural conditions or the experimental condition using fungi growing grapes?

4.      Line 198-200 – Could the results be also interpreted that the infected larvae survived better and/or that the infected parent flies laid more eggs?

Author Response

Wolbachia infections are common in insects, and the questions as to how they persist and spread could be important in both basic and applied science. While Wolbachia infections with little influence on host reproduction (e.g., CI, feminization, or male killing, etc.) can still spread as found in earlier studies from this same group, underlying mechanisms are still unclear in some situations. In the current study, the authors show wAu infected flies have relative fitness advantage (i.e., higher offspring number) when fungi are growing heavily whereas such effect is not pronounced when fungi are less in breeding sites. The observation is quite interesting, and this could be first interpreted that the infection has a fitness advantage when the environment is harsh, which could potentially explain the wide spread of wAu in natural fly populations.  Although the authors could not replicate the results in another experiment that looked at the effect of wAu infection on the life history traits of flies in toxin (OTA) contaminated breeding sites, it might also suggest that there could be other factors that mediated such effect as the effect of fungi (and other things growing on the grapes) could be quite complex.  Perhaps, wAu infection may benefit from some substances produced by fungi or something else. The manuscript is well written overall. The experiments are carefully designed and performed, and the data are analyzed properly with clear presentation.  The application of fluctuating temperature as a norm for baseline experimental environment is notable.  I have no further suggestions except for the minor comments below.

Thanks for these comments. We certainly agree that there is room to explore new aspects here!

1.      Experimental design of fungus exposure – I wonder if it would be possible to control the amount of fungi by inoculating known amount.  Looking at Figure 1, could fungi grow more when there are more infected flies?

It might be possible to undertake such experiments under sterile conditions, but these were not available to us. However we’ve now noted this approach in the discussion. (see line 304). In our case we relied on fungal growth arising by chance to create variability in growth levels. We now also mention that there may be an interaction between fly/Wolbachia/fungal composition but we are unable to separate this (see line 326).

2.      Line 136 – Perhaps authors could add a bit more details as to how growing of wanted species of fungus could be controlled. Are there any complication in interpreting the results linking to the previous study and the OTA experiment in the current study considering many other things could potentially grow in grapes?

We appreciate that there are other microbes growing on grapes and as mentioned in response to the previous comment we now mention the possibility of growing fungus under more controlled conditions. OTA happened to be the most obvious compound worth testing but we acknowledge other possibilities (see lines 322-327).

3.      Line 174 – How the amount of OTA was determined? Is the amount relevant to natural conditions or the experimental condition using fungi growing grapes?

The amount of OTA (0.05 ng /mL) was determined according to Mutlu (2012), which is the amount relevant to natural conditions. A higher concentration (0.5 ng OTA/mL) was also tested to see if there were any effects. This information is now added (line 195).

Mutlu, A. G. Increase in mitochondrial DNA copy number in response to Ochratoxin A and methanol-induced mitochondrial DNA damage in Drosophila. Bull Environ Contam Toxicol 2012, 89, 1129–1132.

4.      Line 198-200 – Could the results be also interpreted that the infected larvae survived better and/or that the infected parent flies laid more eggs?

Yes larval survival and egg numbers could play a role in combination. We now acknowledge this. (see lines 217-218).

Round 2

Reviewer 1 Report

The authors took into account most of the comments made.